# Effect of Microstructure and Tensile Shear Load Characteristics Evaluated by Process Parameters in Friction Stir Lap Welding of Aluminum-Steel with Pipe Shapes

**DOI:** 10.3390/ma15072602

**Published:** 2022-04-01

**Authors:** Leejon Choy, Myungchang Kang, Dongwon Jung

**Affiliations:** 1Graduate School of Convergence Science, Pusan National University, Busan 46241, Korea; leejonee@hanmail.net; 2Faculty of Mechanical, Jeju National University, Jeju 63243, Korea

**Keywords:** dissimilar friction stir welding, tool penetration depth (TPD), microstructure, intermetallic compound (IMC) thickness, process parameter

## Abstract

In recent years, friction stir welding (FSW) of dissimilar materials has become an important issue in lightweight and eco-friendly bonding technology. Although weight reduction of low-rigidity parts has been achieved, the weight reduction has been minimal because high-rigidity parts such as chassis require the use of iron. Considering the difficulty of welding a pipe shape, it is necessary to understand the effect of process parameters on mechanical performance. As a result of the study by various process parameters affecting the joint between aluminum and steel in the shape of a pipe, it can be seen that the tool penetration depth (TPD) has the most important effect on the tensile shear load (TSL). However, the effect of TPD on intermetallic compound (IMC), which has the most important influence on fracture, has not been well established. In this study, the effect of process parameters on IMC thickness and TSL in FSW of A357 cast aluminum and FB590 high tensile steel was investigated to reduce the weight of the torsion beam shaft of an automobile chassis. After the FSWed experiment, measurements were performed using an optical microscope and scanning electron microscopy (SEM) to investigate the microstructure of the weld. The formation of an IMC layer was observed at the interlayer between aluminum and steel. TPD is a major factor in IMC thickness variation, and there is a direct relationship between IMC thickness reduction and TSL increase, except for certain sections where the welding speed (WS) effect is large. Therefore, in order to improve mechanical properties in friction stir lap welding of aluminum and steel for high-rigidity parts, it is necessary to deepen the TPD at a level where flow is dominant rather than heat input.

## 1. Introduction

Recently, energy saving for global environment protection has emerged as an important issue in many industries, including the automobile industry [1,2,3,4]. Replacing steel with aluminum (Al) alloys to improve vehicle performance and fuel economy is a possible alternative. To replace everything made of steel with aluminum is difficult due to the differences of the mechanical performance of the material [5]. Therefore, a potential alternative is the replacement of a part of the steel with the Al alloy. When doing this, it is important to properly join the steel and Al; however, there are several factors that make dissimilar joining of steel and aluminum difficult. In particular, fusion welding typically results in various types of chemical separation, internal defects, and undesirable IMC in Al/steel welding. Conventional welding methods for dissimilar aluminum alloys and steels can lead to the formation of thick intermetallic (IMC) layers, so an alternative is needed [6]. Friction stir welding (FSW) is a stir welding technology developed by TWI in the UK in 1991 [7]. Because FSW is performed below the melting point of the material, it is possible to solve problems such as solidification defects that occur during melt welding, and it is possible to join lightweight alloys and high-strength alloys. A coupled torsion beam axle using high-strength steel (HSS) FB590 material is mounted on the rear of the car to interlink the tire and the body, support the force received from the tire, adjust the rolling angle when cornering, and the absorb vibrations or impacts of the road side [8]. Hot-rolled FB590 is an HSS that can be used in automotive chassis and suspension applications owing to its excellent crash performance, etc. [9]. In a coupled torsion beam axle steel, A357 material was selected to replace the trailing arm part that is not affected by torsion with Al material. In the aerospace and automotive industries, the alloys A356 and A357 are widely used for casting high-strength parts because they provide a combination of high strength while having good casting properties [10,11,12,13]. The FSW of A357 and FB 590 was studied to reduce the weight of automobile parts [14,15,16]. Park et al. [14] reported that by performing dissimilar FSW of 3 mm thick A357 cast Al and FB590 HSS plate, a tensile shear load (TSL) of 72.8% was achieved compared to the Al base material with TSL of 7912 N. The FSW process is mainly divided into four phases, and the normal force is changed in them [17]. During the FSW process, thermal energy is generated by friction between the tool and the workpiece, and plastic deformation of the workpiece occurs [18]. A study was conducted to determine the effects of different process parameters on the heat input and normal force [19,20,21]. The helical shape of the scroll tool shoulder improves the vertical force and is especially suitable for curved joints [22,23]. Process parameters that affect the heat input include the tool geometry and tool-related process parameters (offset and tilt), tool rotational speed (TRS), welding speed (WS), plunge speed, dwell time, plunge depth (PD), and tool penetration depth (TPD). To reduce defects in FSW and increase mechanical strength, the influence of process parameters becomes important after the tool design is complete. In a study on tool-related process parameters, it was reported that there is an optimal point of the normal force as the inclination angle of the tool and the tool offset increase [24,25]. In order to reduce the defects of the FSW joint and increase the mechanical strength, a study was conducted on TRS, WS, plunge speed, dwell time, PD, and TPD, which are non-tool process parameters. Process parameters have been shown to affect the mechanical performance and microstructure [26,27,28,29,30,31,32,33,34,35]. The joint strength has been shown to increase with increasing TPD [31,32,33,34]. In addition, research on various bonding methods between various materials is in progress.

Butts and laps between similar or dissimilar Al alloys [1,17,18,19,20,36], butts between Al alloys and steel [14,37,38,39], and laps [15,16,21,40,41] have been performed. Shamsujjoha et al. [42] studied the effect of tool penetration after friction stir lap welding (FSLW) of AA6061-T6511 and mild steel 1018. Understanding the evolution of microstructures is of considerable interest because the generation of micrograin structures in FSW significantly influences their mechanical properties [24,43,44]. The microstructure is also affected by tool and process parameters. Performing FSW of pipes is hard owing to its complicated geometry, and only a few studies of FSLW have been reported [15,16]. Al and steel materials have near-zero mutual solubility, forming the IMC of Fe_x_Al_y_ [44]. The formation of IMC depends on the solid-state reaction between two substances, Fe and Al [45]. It is proposed that in FSW, the weld strength of aluminum and steel depends on the IMC, which is often formed during the welding operation of dissimilar materials [25,46]. In addition, process parameters directly affect the thickness of IMC [4,26,28,29,39,40,47,48,49]. Hussein et al. [4] presented a cause-effect diagram for IMC thickness formation. Many studies have been conducted to determine the relationship between process parameters and IMC thickness [25,26,29,33,50,51,52], while many studies have also been conducted to determine the relationship between the TSL and IMC thickness [53,54,55]. The latest research trends on IMC thickness and pipes are as follows. Aghajani Derazkola et al. [56] reported that the water-cooled sample with optimal IMC layer thickness showed the highest strength in friction stir joints of AA3003 and A441 AISI steels. A lower cooling rate allows thickening of IMCs and provides more brittleness to decrease the strength of FSW. Mortello et al. [57] obtained best results at feed rates ranging from 1.3 to 1.9 mm/s and rotational speeds of approximately 700 to 800 rpm for friction stir lap joints of AA5083 H111 and S355J2 grades DH36 structural steel. As the IMC thickness increases, the shear force decreases. Pankaj et al. [58] showed that in butt friction stir joints of DH36 steel and AA6061, the IMC thickness decreased with increasing tool offset and increased with increasing tool offset from 0.4 to 1.5 mm. The tool offset was observed to decrease with further increase from 1.5 mm to 2.5 mm. Abd Elnabi et al. [59] reported that the shear tensile strength increased as the tool pin length increased, and there was no change in bonding efficiency at IMC thicknesses of 7.5 μm or more. As a recent research of pipe, Choy et al. [16] used the definitive screening design method in the FSLW of aluminum and steel pipe as a previous study in this paper. PD and TPD are the most important factors influencing TSL, and there is no interaction between PD and TPD for the first time. Sabry et al. [60] reported FSW and submerged friction stir welding (UWFSW) on AA6061 pipe with a diameter of 30 mm and a wall thickness of 2, 3, and 4 mm. Using the full factorial analysis method, it was stated that the higher the TRS, the higher the UTS increased as the WS decreased. Abdullah et al. [61] investigated the rotational and linear velocities of tools in friction stir lap joints of AA5086 and C12200 copper alloy pipes and the microstructure, macrostructure, and mechanical properties of tool joints. Cylindrical pin tools at speed ratios below 10 rev/mm and conical pin tools in the 5–15 rev/mm speed ratio range have been shown to produce tunnel defects in the agitation zone. It is important to study process parameters because the type of IMC, the bonding mechanisms (such as hook joints, micro, and macro structures), and the IMC thickness (related on the axial pressure and heat generation of the weld zone). IMC thickness depends on the relationship between input heat and material flow. In general, increasing TRS increases IMC thickness and increasing WS decreases IMC thickness. In this study, TPD was assumed as the factor that has the greatest influence on material flow among process variables. So far, most of the studies have been mainly conducted on FSW joints to determine the influence of process parameters related to the plate shape. Pipe geometry studies that focus on the influence of process parameters have been mainly conducted on similar butt joints. Even in the study of pipe shape, only a few studies that focus on the effects of microstructures on the FSLW of AL and steel have been reported. In addition, there is little literature describing the effect of the microstructure on TPD, other than TRS and WS. Therefore, in this study, we aim to study the microstructure, as well as the IMC thickness and TSL characteristics according to TPD in pipe-shaped Al and steel FSLW.

## 2. Experimental Preparation and Methods

### 2.1. Materials and Tools

The pipes used in this experiment were A357 cast Al and FB590 HSS. The chemical composition of each material is shown in Table 1 [14]. The test specimen, i.e., A357 cast aluminum, was manufactured with an outer diameter of Φ 111 mm, a length of 155 mm, a thickness of 3 mm in the joint, and a thickness of 6 mm in an unbonded part.

FB590 HSS was manufactured with an outer diameter of Φ 105 mm, a length of 110 mm, and a thickness of 3 mm. Figure 1 is a photograph showing the experimental equipment, tool, and TPD. As shown in Figure 1a, the experimental equipment consists of a Winxen milling equipment that supplies the rotational force of the spindle up to 2000 rpm, a chuck fixing both sides of the pipe, and a fixing jig equipped with a bearing to support the pipe. Figure 1b shows the tool used in the FSW processing and the enlarged picture of the scroll shape of the shoulder. Figure 1c shows the change in TPD according to the change of tool, with pin lengths of 3.0, 3.5, and 4 mm in the boundary layer between 3 mm thick aluminum and steel, in order to investigate the change in IMC thickness according to TPD. To rule out the effect of the PD, the PD was fixed at 0.0 mm. Dotted box arrows indicate increases in TPD. The material of the FSW tool was manufactured using W–Ni–Fe alloy, which is a type of heavy alloy. The tool’s pin was processed into a threaded shape with a cylindrical tape, and the tool’s shoulder was processed into a parallel scroll shape to increase the *z*-axis vertical force, improving the frictional heat and stirring during the FSW joining process [14,23].

The shoulder, pin root, and pin diameter of the tool are 10, 5, and 4 mm, respectively. To determine the effects of TPD, three types of tools with pin lengths of 3.0, 3.5, and 4.0 mm were used.

### 2.2. Experimental Methods

Figure 2 is a schematic diagram of the FSLW experimental process to investigate the effect of the IMC thickness and TSL according to different process parameters. Figure 2a, b show the definition of process parameters on the experiment. Figure 2a shows the TRS at which the main shaft rotates at a constant speed, the WS at which the tool advances the workpiece for joining the workpiece, the plunge speed descending at a constant speed relative to the TPD, and TPD at a constant plunge speed. After the tool descends to destination point, the descent is stopped, and the definition of dwell time is shown while the tool rotates in a fixed position for a certain period of time. For tools pre-selected by the PD and TPD in Figure 2b, the PD to which the top of the tool pin penetrates the outer diameter of the pipe and the TPD through which the tool pin penetrates the interlayer between steel and Al are shown. Figure 2c shows the thickness of the IMC layer, which is required to investigate the effect of process parameters on the microstructure. Figure 2d is shown to investigate the effect of process parameters on the magnitude of TSL.

According to Figure 2, the experimental method is as follows. After the experiment was performed according to the number of levels of the process parameter based on the set experiment, the change in the thickness of the IMC layer affected by each process parameter was measured. After the measurement, the effects of changes in the TPD and TRS were analyzed.

Shamsujjoha et al. [42] reported a better weld strength when the Al plate was placed along the advancing face on the steel plate. The two materials were washed with ethanol and then installed using the lap method, with A390 cast Al on the top side and FB590 HSS on the bottom side; the lap width was about 30 mm, as illustrated in Figure 1.

Table 2 welding parameters for experiment. Group A is experiment no. 1, no. 2, and no. 3, and group B is experiment no. 4, no. 5, and no. 6.

The sequence of experiments and the parameter values were selected to investigate the process parameters affecting the IMC thickness and TSL.

Because the PD and the TPD are affected by the pin length of the tool, the PD was fixed at 0.0 mm to prevent mutual interaction and to only investigate the effect of TPD, such that it is not affected by the PD. Tool pin lengths of 3.0, 3.5, and 4.0 mm were selected and used for the TPD experiment. Experimental conditions for the remaining parameters were selected as follows: TRS of 1700–1900 rpm; WS of 0.1–0.2 rpm; and a TPD of 0.0 mm, 0.5 mm, and 1.0 mm. The experiment was divided into two groups to investigate the effect of WS and TPD on the IMC thickness and TSL. The process parameter of one group is 0.1 rpm WS, and the other group has a WS value of 0.15 rpm or more. Figure 1c shows the change according to the TPD and indicates the position of the tool penetrating the steel surface through the boundary layer between A357 and FB590 when PD = 0 and TPD is 0.0, 0.5, and 1.0 mm. In Figure 1c, the box arrows indicate the direction in which the TPD increases. It was concluded that the TPD plays an important role in determining the weld strength [35].

To investigate the effect of the WS and TRS on the IMC thickness and TSL, the same experimental results were divided into two groups based on TRS. One group of reclassified results of experiment was fixed at a WS of 0.1 rpm, and the other group was a WS of 0.15 rpm or more.

Experiments were performed on an FSW machine (Winxen, 22 kN). Pipes are installed on the chuck and fixture, and were lap-bonded, as shown in Figure 1a. Welding is done in quadrants, i.e., 90°, instead of full rotation. To observe the structure of the junction, a rectangular sample was prepared. The tensile test specimen was manufactured according to the ASTM E8 standard, and TSL was measured using a tensile tester (AGX-X, Shimadzu, Kyoto, Japan). The size of the specimen for microstructure measurement was selected to be 20 mm wide and 10 mm long, for hot mounting The specimen was fabricated using EDM wire (NW570Ⅱ, Doosan, Changwon, Korea) along a line perpendicular to the welding direction. The sample that was used to observe the microstructure of the FSW section was polished with silica sandpaper in a disk grinder and then polished with a diamond cloth. Weld textures and bonding interfaces between Al and steel were studied using optical microscopy (KH-8700, HIROX, Tokyo, Japan) and scanning electron microscopy (JSM-6490, JEOL, Tokyo, Japan).

## 3. Results and Discussion

### 3.1. Microstructure Characteristics of Friction Stir Welding (FSW) Joint

Figure 3 shows the optical macrostructure and microstructure of the FSW specimen. Figure 3a shows the macrostructure of the cross section of the FSWed joint according to the experimental conditions. The macrostructure in Figure 3a is enlarged. The TMAZ is shown in Figure 3b and SZ in Figure 3c. The HAZ is shown in Figure 3d, the Al base material (BM) is shown in Figure 3e, and the IMC region is shown in Figure 3f, respectively. In Figure 3a, the broken line of the magnification indicates the shape of the tool pin, and the arrow spacing of the PD indicates the spacing between the outer diameter of A390 cast Al and the shoulder of the tool. The arrow spacing of the TPD indicates the gap between the tip of the tool pin and the interlayer at which the inner diameter of A390 Al and the outer diameter of FB590 HSS meet. The cause of the cavity in Figure 3a is the geometric difference between the pipe and the plate. Akbari et al. [13] found that the contact characteristics of the tool and the workpiece were different due to the small radius of curvature of the pipe. At a PD of 0.1 mm, due to the pipe curvature, the FSW shoulder partially contacts the pipe and only the inner part makes the shoulder part contact the pipe, forming a tunnel cavity. Another cause of large cavity is a lack of material. (1) The upper aluminum material under the influence of process variables is discharged to the outside in the form of flash. (2) The upper aluminum material moves to fill the gap (gap) on the junction interface of aluminum and steel. The lack of material due to the above causes created a large internal cavity.

After FSW, there is an SZ around the center of the joint. There is a TMAZ in which magnitude of grains are increased by plastic flow on the outside of the SZ, and there is an HAZ that is heat-affected but has no plastic deformation on the outside of TMAZ. These zones were observed to have a wider region width than the retreat side on the tool’s advance side (AS). To form a bond as the tool is traversed, material flows around the tool in a complex flow pattern depending on the tool geometry and process parameters, such as TRS and WS. Microstructural evolution depends on the change of the welding parameters. A suitable selection of the process parameters results in good material mixing at the joint and a sound weld.

The SZ in Figure 3c produces a fine grain structure, whereas the TMAZ in Figure 3b has a long grain structure. The SZ region of Figure 3c shows finer particles compared to the TMAZ region and the HAZ region of Figure 3d. In general, the SZ region is also referred to as a “nugget region” and is believed to be formed by dynamic recrystallization. The SZ region has a wider width at the top and decreases in the thickness direction. This is because of the difference in diameter between the shoulder and the pin. In the SZ, a finer grain was formed compared to other regions. Dynamic recrystallization produced skewed and elongated grains in TMAZ and fine grains in SZ. The grain boundary orientation is different in all three regions. The size of the SZ region in Figure 3c is smaller than that of the BM, and the crystalline grains of the BM disappear. This is in agreement with the results of Mahto et al. [21], who reported that new crystals were formed and dynamic recrystallization occurred. The microstructure of TMAZ is shown in Figure 3b. Located between HAZ and SZ, TMAZ is characterized by a highly deformed structure. It shows an elongated non-metallic particle morphology with sub-grain coarsening. The grain structure of the HAZ region, which was not mechanically perturbed by FSW, was generated by static recrystallization and was similar to the base metal in Figure 3e HAZ is much wider at the top surface in contact with the shoulder and is narrow as the pin diameter decreases. In Su et al. [62], the grain size increases in the order of SZ, TMAZ, HAZ, and BM. Therefore, the microstructure was recrystallized in the FSWed region.

### 3.2. IMC Thickness Characteristics of FSW Joints

In order to investigate the change in the IMC thickness due to the TPD, the PD is fixed at 0.0 mm, and changes in the remaining process parameters are considered. First, the effect on the IMC thickness during FSW according to the TPD of group A with a WS of 0.1 rpm is investigated. Figure 4 shows the change in the IMC thickness during FSW according to the TPD in group A as an SEM photograph. Figure 4a shows the IMC average thickness when the TPD is 0.0 mm in experiment no. 1, Figure 4b shows the IMC average thickness when the TPD is 0.5 mm in experiment no. 2, and Figure 4c shows the IMC average thickness at a TPD of 1.0 mm in experiment no. 3. At a TPD of 0.0 mm, the IMC average thickness is 5.1 μm; at a TPD of 0.5 mm, the IMC average thickness is 3.9 μm; and at a TPD of 1.0 mm, the IMC average thickness is 3.1 μm. It can be seen that the IMC average thickness decreases as the TPD increases. Next, the effect of the TPD on the IMC average thickness was investigated in group B with a WS of 0.15 to 0.2 rpm during FSW. Figure 5 shows the change in the IMC average thickness during FSW according to the TPD in group B as an SEM photograph. Figure 5a–c shows the IMC average thickness of experiment no. 4 with a TPD of 0.0 mm, experiment no. 5 with TPD of 0.5 mm, and an experiment no. 6 with a TPD of 1.0 mm. When the TPD is 0.0 mm, the IMC average thickness is 9.05 μm; at a TPD of 0.5 mm, the IMC average thickness is 6.43 μm; and at a TPD of 1.0 mm, the IMC average thickness is 5.07 μm. As in group A where the WS is 0.1 rpm, it can be seen that the IMC average thickness decreases as the TPD increases.

Mahto et al. [28] reported that the higher the heat input, the thicker the IMC layer was in the FSLW of AA 6061-T6 and AISI 304 stainless steel, each having a thickness of 1 mm. According to Ogura et al. [63], the nature of the IMC formed during welding is determined by the combined influence of base materials stirring and thermal conditions. As the TPD increases, the contact area between the tool and the material increases and the heat input tends to increase. Another trend is that the flow of material will increase due to agitation as the TPD increases. Therefore, it can be deduced that the thickness of the IMC layer decreases because the effect of the material flow due to agitation is greater than the increase in the heat input due to the increase of the TPD.

### 3.3. Characteristics of IMC Thickness and Tensile Shear Load (TSL) According to the Tool Penetration Depth (TPD)

Table 3 summarizes the IMC thickness and TSL according to the change in TPD and WSs of 0.1 rpm and 0.15 rpm or more.

Figure 6 compares the differences between the two groups with changes in the IMC average thickness according to TPD in group A with a WS of 0.1 rpm and group B with a WS of 0.15 to 0.2 rpm. In Figure 6, group A of experiments no. 1, no. 2, and no. 3 is indicated by a solid line to represent the IMC average thickness change according to the TPD and a WS of 0.1 rpm. It shows a gradual decrease in the IMC average thickness as TPD increases from 0.0 mm to 1.0 mm. In Figure 6, group B of experiments no. 4, no. 5, and no. 6 are indicated by broken lines to represent the IMC average thickness change at WS values of 0.15 and 0.2 rpm. It shows a gradual decrease in the IMC average thickness as the TPD increases from 0.0 mm to 1.0 mm. It can be seen that the IMC average thickness gradually decreased as TPD increased from 0.0 mm to 1.0 mm, regardless of the influence of WS and other process parameters.

In addition, it can be seen that the IMC average thickness increases as the WS increases at the same TPD. This is consistent with the experimental results reported by Shen et al. [34]. IMC at the Fe–Al interface decreases with increasing TPD during the FSLW of 2.2-mm-thick AA5754 Al and 2.5-mm-thick DP600 dual-phase steel. However, their results did not provide quantitative results for IMC thickness increase. Mahto et al. [35] performed FSLW between 0.0 mm and 0.3 mm with a pin depth (PLD) of 0.1 mm spacing on 1-mm-thick dissimilar materials AA6061-T6 and AISI304. It is said that the IMC thickness increases as the PLD increases. This is the result of changing the TPD in the narrow range of 0–0.3 mm, and it appears to be different from the current wide region, where the TPD is changed between 0.0 and 1.0 mm. In the study reported by Liu et al. [25], it was proposed that the IMC thickness decreased with the increase of the WS and the increase of the tool offset. In this study, the influence of TPD was dominant, showing the opposite trend. From the experimental results, as the TPD increases, the material fluidity increases and the IMC thickness decreases. This is because the effect of TPD is more dominant on material flow than on heat input.

Figure 7 shows graphs of the variation of the TSL with TPD.

In Figure 7, group A of experiments no. 1, no. 2, and no. 3 is indicated by a dash-dotted line to represent the changes in the TSL with the TSL and a WS value of 0.1 rpm. As TPL increases from 0.0 mm to 1.0 mm, it can be seen that the TSL decreases up to a TPD value of 0.5 mm, and TPD increases above 0.5 mm. In Figure 7, group B of experiments no. 4, no. 5, and no. 6 are indicated by dash-double dotted lines to represent the changes in the TSL for WS values of 0.15 and 0. 2 rpm. It shows a gradual increase in the IMC average thickness as the TPD increases from 0.0 mm to 1.0 mm. It can be seen that the IMC average thickness gradually decreased as TPD increased from 0.0 mm to 1.0 mm, regardless of the influence of WS and other process parameters. As TPD increases, TSL decreases to TPD = 0.5 mm at a WS of 0.1 rpm. At the WS of 0.1 rpm, the TPD increases over 0.5 mm, and at the WS of 0.15 and 0.2 rpm, the TPD increases over the entire region. In the study of TSL according to TPD, Lee et al. [52] reported that the TSL increased as the TPD increased, but this result is due to the difference between FSSW and this study, which has a large initial thermal effect. In order to increase the TSL according to TPD, a WS above a certain speed was not proposed. Shen et al. [34] reported that the TSL decreased and increased with increasing TPD. Contrary to the results of this study, this results from an inability to distinguish between the mutual effects of TPD and PD.

In Figure 6 and Figure 7, as TPD increases at a WS of 0.1 rpm, the IMC average thickness decreases and TSL decreases until TPD = 0.5 mm. In Figure 8, it can be seen that the IMC average thickness decreases and the TSL increases at TPD = 0.5 mm or more as the TPD increases at the WS of 0.1 rpm. When WS varies from 0.15–0.2 rpm, the IMC average thickness decreases and TSL increases as TPD increases in all regions. Picot et al. [54] used the process parameters of the rotational speed and translational speed of the tool in FSLW of stainless steel 316L and Al alloy 5083 and reported that the TSL decreased with increasing IMC average thickness. Helal et al. [55] used the process parameters of TRS, WS, and tool offset in FSLW of 6061-T6 Al alloy and ultra-low carbon steel and reported that as the IMC average thickness increases, TSL decreases. Kimapong et al. [53] reported that TSL decreased with increasing IMC average thickness using welding parameters such as rotational speed, traversing speed of 0.4 to 1.43 mm/s, and a pin depth in FSLW of A5083 Al alloy and SS400 steel. In previous studies, it was reported that the TSL increased with decreasing IMC average thickness. As WS varies from 0.15–0.2 rpm, it was consistent with the research results of previous researchers. However, at the low WS of 0.1 rpm, for a TPD value up to 0.5 mm, the IMC average thickness decreased as the TPD increased, but the TSL decreased.

### 3.4. Characteristics of IMC Thickness and TSL According to TRS

Table 4 summarizes the IMC thickness and TSL according to the change in the TPD and for WS values of 0.1 rpm and 0.15 rpm or more. Figure 8 shows the relationship between the IMC thickness and TSL according to the change in the TRS.

Figure 8 compares the differences between the two groups in terms of the change in the IMC thickness according to TRS in group A with a WS of 0.1 rpm and group B with a WS of 0.15 to 0.2 rpm. In Figure 8, group A of experiments no. 1, no. 2, and no. 3 is indicated by a solid line to represent the changes in the IMC average thickness change according to TRS and a WS of 0.1 rpm. As TRS increases, it can be seen that the IMC average thickness increases up to TRS of 1800 mm and decreases when TRS exceeds 1800 mm. In group B of experiments no. 4, no. 5, and no. 6 in Figure 8, at WS values of 0.15 and 0.2 rpm, the variation in the IMC average thickness according to TRS is indicated by broken lines. It shows a gradual increase in the IMC average thickness as TRS increases from 1700–1900 rpm. When TRS is increased from 1700–1900 rpm, the tendency of the IMC average thickness change is different according to the effect of WS. As TRS increased, the IMC average thickness decreased for a WS value of 0.1 rpm and when TRS of 1800 rpm or more. When WS was 0.1 rpm, TRS was 1800 rpm or less, and when WS was 0.15 and 0.2 rpm, it showed an increase over the entire range of TRS.

In the study by Kundu et al. [51], the IMC thickness increased with increasing TRS. These results do not indicate conditions for WS above a certain speed. Wan et al. [25] and Das et al. [51] simultaneously studied the increase in the IMC thickness according to the increase of TRS, and the decrease in the IMC thickness according to the increase in WS. However, their study also did not report that the IMC thickness increased with increasing TRS at WS above a certain speed.

Figure 9 is a graph showing the relationship between the TSL and TRS. In the figure, group A of experiments no. 2, no. 1, and no. 3 is indicated by a dash-dotted line to represent the TSL change according to TRS at WS of 0.1 rpm.

As TRS increases from 1700 rpm to 1900 rpm, it can be seen that the TSL increases up to a TRS of 1800 rpm, and TSL decreases for a TRS of 1800 or more. In Figure 9, group B of experiments no. 4, no. 5, and no. 6 is indicated by dash-double dotted lines to represent the change in the TSL according to TRS at WS values of 0.15 and 0.2 rpm. As the TRS increases from 1700 rpm to 1900 rpm, the TSL shows a gradual decrease. When TRS is increased from 1700 to 1900 rpm, the tendency for the change in TSL differs according to the effect of WS. As TRS increased at the WS of 0.1 rpm, the TSL increased up to TRS of 1800 rpm. It shows that as TRS increases at the WS of 0.1 rpm, the TSL decreases over the entire range of TRS when TRS is over 1800 rpm and at WS values of 0.15 and 0.2 rpm.

With respect to the TSL and TRS, Kimapong et al. [53] stated that the TSL decreased with increasing TRS. Their study also did not consider the effect of WS on specific regions. In this study, as in the study by Kimapong et al. [53], the TSL decreased as TRS increased at a WS of 0.15 to 0.2 rpm. However, it was found that the TSL increased as TRS increased at the WS of 0.1 rpm and when TRS was 1800 rpm or less.

In Figure 8 and Figure 9, at a WS of 0.1 rpm and below TRS of 1800 rpm, the IMC average thickness increases and the TSL increases as TRS increases. At a WS of 0.1 rpm, if TRS is 1800 rpm or more, the IMC average thickness decreases and the TSL decreases as TRS increases. When WS ranges from 0.15 to 0.2 rpm, the IMC average thickness decreases and TSL decreases as TRS increases in all regions.

Picot et al. [54] used the process parameters of the rotational speed and translational speed of the tool in the FSLW of stainless steel 316L and Al alloy 5083 and reported that the shear strength decreased with increasing IMC average thickness. Helal et al. [55] used the process parameters of TRS, WS, and tool offset in FSLW of 6061-T6 Al alloy and ultra-low carbon steel and reported that as the IMC average thickness increases, TSL decreases. Kimapong et al. [53] reported that the shear strength decreased with increasing IMC thickness using welding parameters such as the rotational speed, traversing speed of 0.4 to 1.43 mm/s, and a pin depth in FSLW of A5083 Al alloy and SS400 steel.

In previous studies, it was reported that the TSL increased with decreasing IMC average thickness. At the WS of 0.15–0.2 rpm, it was consistent with the research results of previous researchers. However, as TRS increased in all regions of TRS at a low WS of 0.1 rpm, the TSL increased as the IMC average thickness increased, and the TSL decreased as the IMC average thickness decreased. Many studies have been done to determine the relationship between the other process parameters and the IMC thickness. A study by Liu et al. [25] indicated that the IMC thickness decreased as the WS increased and the tool offset increased. However, their study results also reported a decrease in the IMC thickness with an increase in the tool offset but did not provide a condition for the WS over a specific speed. Dehghani et al. [29] showed that the IMC thickness increased with increasing PD.

In addition, when the WS was constant, the effect of the TPD was greater than the rotational speed of the tool. This is consistent with the fact that according to the results of this study, the effect of the TPD is dominant. Many studies have also been done to assess the maximum size criterion of the IMC thickness for strength [25,36,46,47]. Jamshidi et al. [36] recommended a critically reactive layer of IMC as thin as 1–2 µm for strong bond strength, as a thicker layer of IMC lowers the TSL and increases the hardness while reducing the bond ductility. However, even if it is thinner than this dimension, the mechanical properties of the joint are said to be improved. Liu et al. [25] reported that the maximum thickness of the IMC at the butt joint of AA6061-T6 and 780/800 trip steel was 1 μm, which was low enough for good weld strength. Movahedi et al. [46] reported that the presence of IMC in FSLW between Al-5083 and St-12 improves the weld strength when the thickness is less than 2 μm. Meanwhile, Sepold et al. [47] showed that excellent mechanical properties could be obtained by minimizing the thickness of the IMC layer to 10 μm or less. In this study, a region with an average IMC thickness of 3.42 to 9.05 μm was obtained over the entire region. In general, the type, size, and amount of IMC formation depends on the input heat, which is controlled by the welding process parameters. An increase in rotational speed and friction time is expected to increase the heat input. Jamshidi et al. [36] also reported that a larger amount of IMC should be formed because the amount of heat input increases with TPD. Dehghani et al. [29] reported that as the PD increased, the interaction between the shoulder and the workpiece increased and the frictional heat was generated more so that the material was sufficiently plasticized, and the material movement around the tool pin was promoted, resulting in higher heat generation.

This study differs from previous studies in that the IMC thickness increases because of the increase in heat input as the PD and TPD increase. It is estimated that the increase in the friction amount with the increase of the TPD leads to an increase in the amount of heat input. However, in the mechanism according to TPD, flow phenomena other than the heat input appear to act as a more dominant factor than the increase in heat input according to TPD. Therefore, it can be estimated that the thickness of the IMC is reduced. Therefore, these results differ from the experimental results of the one factor at a time, in which the effects of PD and TPD were not tested together. Previous studies have shown that the thickness of the IMC increases with increasing TRS. In this study, it was confirmed that the thickness of IMC decreased as TRS increased at a WS value of 0.1 rpm and a TRS of 1800 rpm or higher. In a previous study, the IMC thickness decreased with increasing WS. In this study, the IMC thickness increased with increasing WS. Based on the results obtained, it can be seen that the effect of TPD is a significant factor in the reduction of the IMC thickness.

## 4. Conclusions

In this study, the following major results were obtained by studying the characteristics of the TSL and IMC thickness according to TPD and TRS in FSLW for dissimilar Al and steel pipe.

As a result of observing the structural change of the cross section after FSW, the average thickness of the IMC layer at the interface between aluminum and steel in the microstructure ranged from 3.4 to 9.05 μm, and the formation of the IMC layer was observed.As TPD increased, the average thickness of IMC decreased and TSL increased in the entire experimental area except for the area where TSL decreased at TPD = 0.5 mm or less when WS was 0.1 rpm.When WS was 0.1 rpm, TSL increased with increasing IMC thickness at TRS below 1800 rpm, and TSL decreased with decreasing IMC thickness at TRS above 1800 rpm. TSL decreased with increasing IMC thickness as TRS increased when WS was above 0.15 rpm.TPD is the dominant factor in the change in size of IMC thickness, and IMC thickness decreases with increasing TPD. TSL increases with decreasing IMC thickness. However, at low WS, when TRS is increased, TSL is affected by TPD, and when TPD increased, TSL was affected by TRS.

These results indicate that TPD, IMC thickness, and TSL were directly related, except for in certain areas with low WS. It is assumed that the increase or decrease of TSL in this region is determined by the relative superiority of heat input and material flow.

Therefore, in order to increase the mechanical properties in the FSLW of aluminum and steel for high-rigidity parts, the TPD should be deepened in the weld area where the WS is not high. A suitable TPD should therefore be considered along with tool wear, and continuous material development and the selection of an optimized TPD will play an important role in future dissimilar FSLW.

## Figures and Tables

**Figure 1 materials-15-02602-f001:**
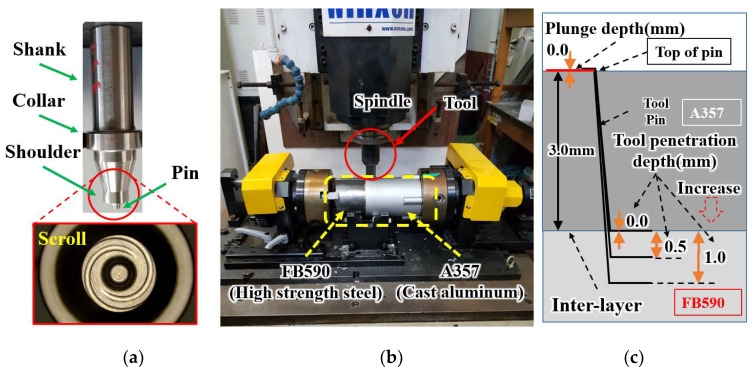
Overall configuration diagram of FSW experiment. (**a**) tool and close view of shoulder; (**b**) photograph of experimental equipment; and (**c**) tool penetration depth (TPD) [16].

**Figure 2 materials-15-02602-f002:**
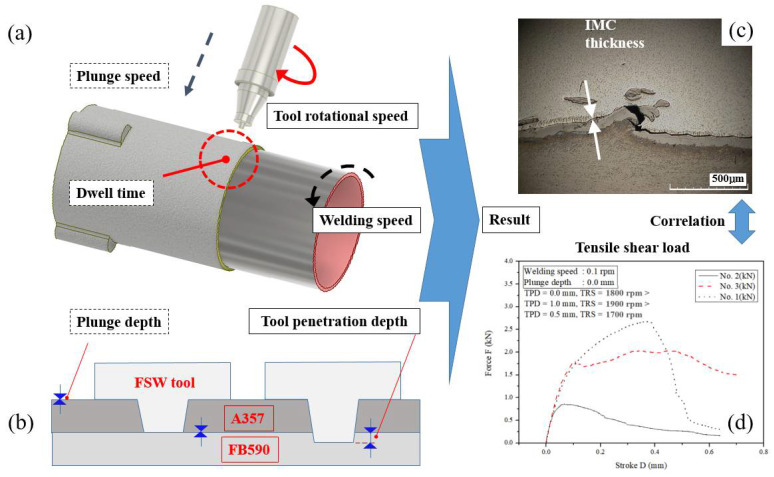
Schematic illustration of FSLW experimental process on IMC thickness and tensile shear load (TSL) by process parameters; (**a**) process parameter; (**b**) plunge depth and TPD; (**c**) IMC thickness; and (**d**) TSL.

**Figure 3 materials-15-02602-f003:**
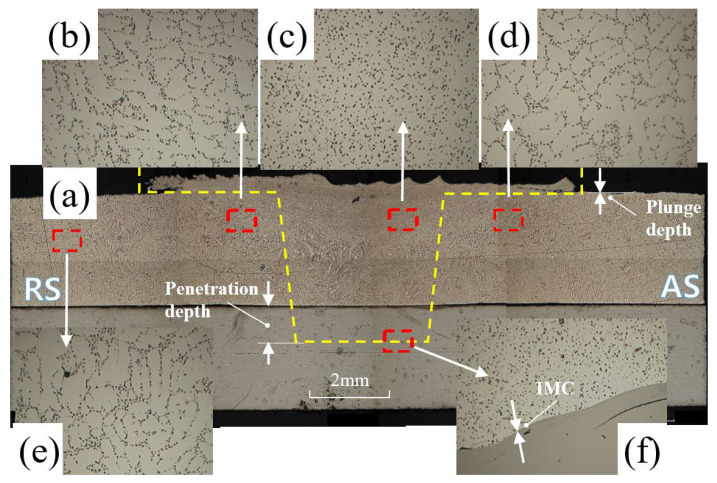
Optical micrographs of the FSW A357 cast alloy and FB590 HSS. (**a**) Overall cross-sectional macrostructure; (**b**) microstructure of the TMAZ; (**c**) microstructure of the SZ; (**d**) microstructure of the HAZ; (**e**) microstructure of the Albase metal; and (**f**) microstructure of the IMC [16].

**Figure 4 materials-15-02602-f004:**
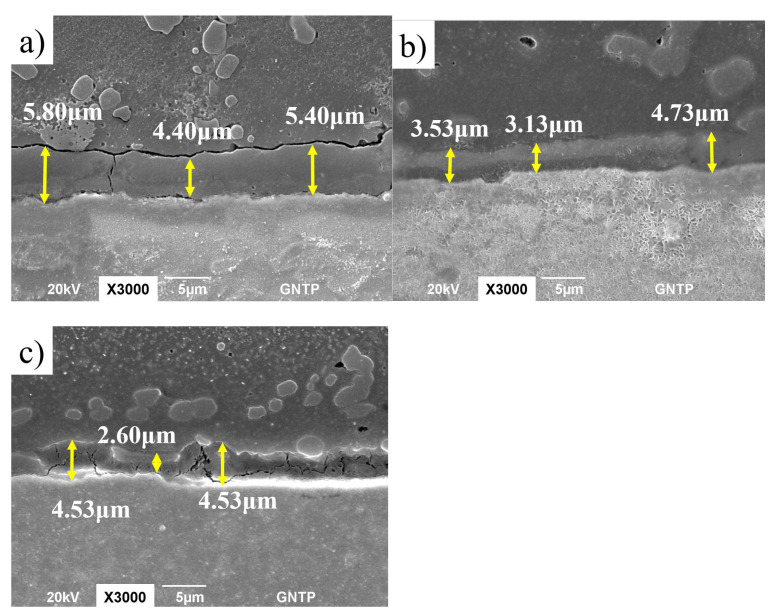
SEM pictures according to TPD of experiment group A. (**a**) TPD = 0.0 mm; (**b**) TPD = 0.5 mm; and (**c**) TPD = 1.0 mm.

**Figure 5 materials-15-02602-f005:**
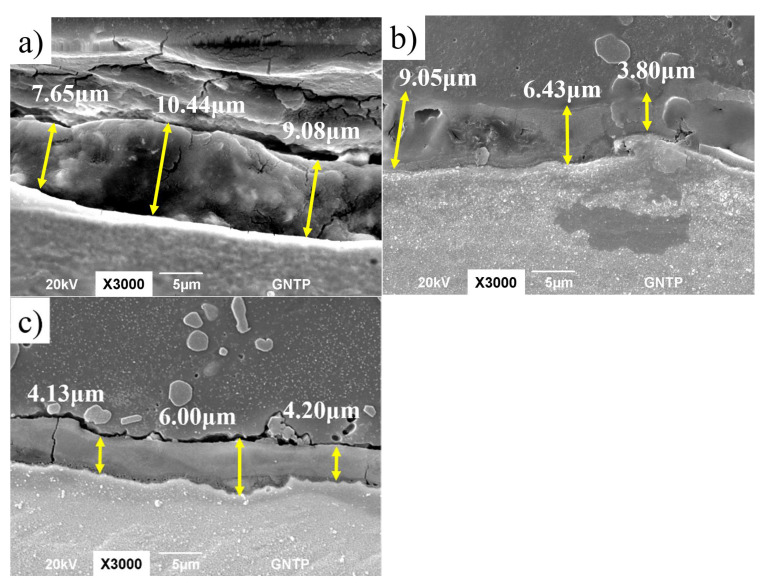
SEM pictures according to TPD of experiment group B. (**a**) TPD = 0.0 mm; (**b**) TPD = 0.5 mm; and (**c**) TPD = 1.0 mm.

**Figure 6 materials-15-02602-f006:**
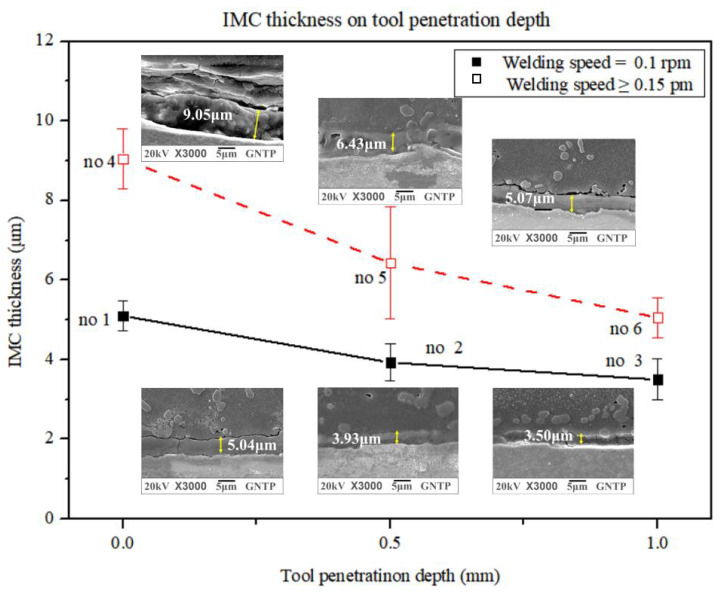
IMC thickness comparison according to TPD for experiment groups A and B.

**Figure 7 materials-15-02602-f007:**
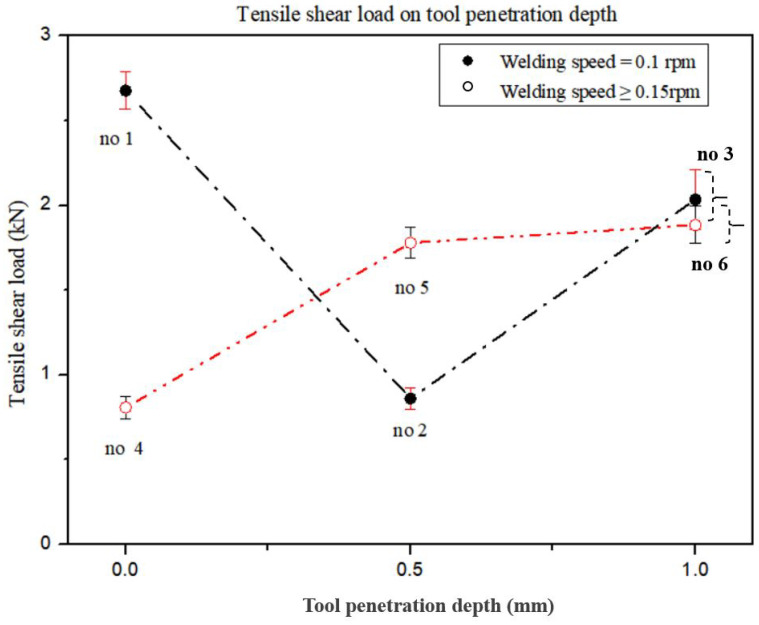
Comparison of TSL with TPD for experiment groups A and B [16].

**Figure 8 materials-15-02602-f008:**
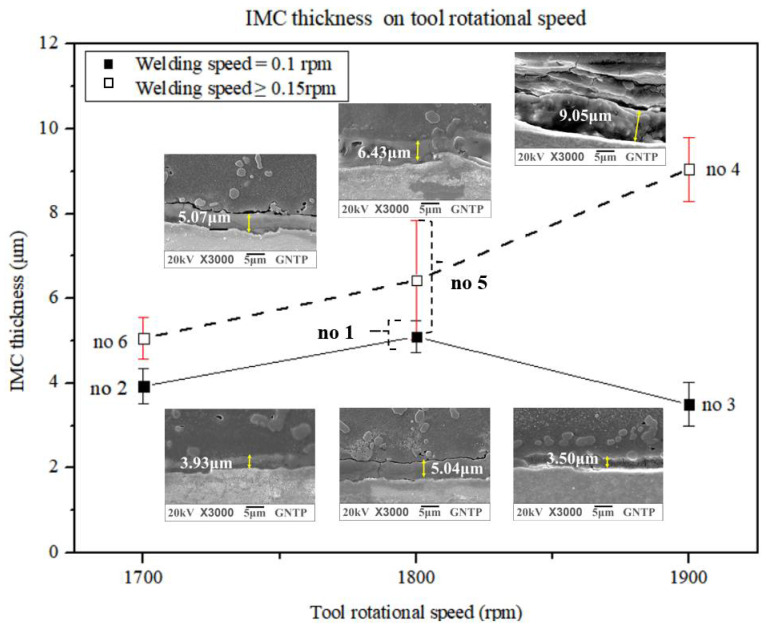
Comparison of IMC thickness for different TRSs of experiment groups A and B.

**Figure 9 materials-15-02602-f009:**
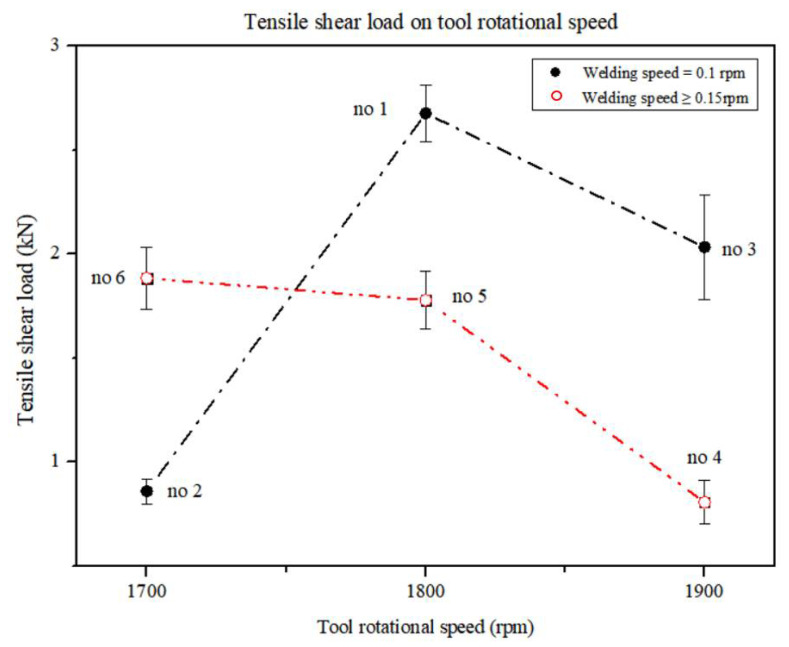
Comparison of TSL with TRS for experiment groups A and B [16].

**Table 1 materials-15-02602-t001:** The chemical composition of base materials [14].

Material	C	Si	Mn	P	S	Cr	Ni
FB590	0.076	0.094	1.472	0.013	0.001	0.019	0.008
**Material**	**Si**	**Mg**	**Cu**	**Zn**	**Fe**	**Mn**	**Ti**
A357	6.937	0.507	0.034	0.017	0.181	0.007	0.116

**Table 2 materials-15-02602-t002:** Welding parameters for experiment.

Experiment No.	Welding Speed (rpm)	Tool Penetration Depth (mm)	Tool Rotational Speed (rpm)
1	0.10	0.0	1800
2	0.10	0.5	1700
3	0.10	1.0	1900
4	0.20	0.0	1900
5	0.15	0.5	1800
6	0.20	1.0	1700

**Table 3 materials-15-02602-t003:** IMC thickness range and IMC average thickness and TSL on TPD.

Experiment No.	1	2	3	4	5	6
Tool penetration depth (mm)	0.0	0.5	1.0	0.0	0.5	1.0
IMC thickness range (µm)	5.80–4.40	4.73–3.13	4.53–2.60	10.44–7.65	9.05–3.80	6.00–4.13
IMC average thickness (µm)	5.04	3.93	3.56	9.05	6.43	5.07
Tensile shear load (N) [16]	2677.21	860.50	2034.83	807.35	1779.25	1984.72

**Table 4 materials-15-02602-t004:** IMC thickness range and thickness average and TSL on TRS.

Experiment No.	2	1	3	6	5	4
Tool rotational speed (rpm)	1700	1800	1900	1700	1800	1900
IMC thickness range (µm)	4.73–3.13	5.80–4.40	4.53–2.60	6.00–4.13	9.05–3.80	10.44–7.65
IMC average thickness (µm)	3.93	5.04	3.56	5.07	6.43	9.05
Tensile shear load (N) [16]	860.50	2677.21	2034.83	1984.72	1779.25	807.35

## Data Availability

Not applicable.

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
