# Peer review of "Effect of Microstructure and Tensile Shear Load Characteristics Evaluated by Process Parameters in Friction Stir Lap Welding of Aluminum-Steel with Pipe Shapes"

_materials, 2022, doi:10.3390/ma15072602_

Round 1

Reviewer 1 Report

In my opinion, this paper contains serious flaws and it requires a lot of work to be published. My most crucial remarks are listed below.

  1. The introduction part in some places is very hard to follow, e.g. “For A357 cast aluminum raw material, the tensile shear strength is 7,912N” (lines 68-69) – what this information has to do with the rest of the text? Would not be better to give this value in MPa? And where is a reference? Another example: “The connection of an automobile structure is mainly a lap connection, and in Al-steel bonding, the process parameters significantly influence the microstructure of the stir zone (SZ), thermos-mechanically affected zone (TMAZ), and 93 heat-affected zone (HAZ) recrystallization [29, 39, 40].” (lines 91-94). First, it is well-known fact that properties of the mentioned zones depend on process parameters, and for it is the only information which this sentence contains it can be removed from the introduction part and it will change nothing. Second, it is “thermo-mechanically” not “thermos-mechanically”. Third, what does the word “recrystallization” do at the end of the sentence? What is the point of it? The introduction part is relatively long but e.g. lines 96-107 bring nothing new to the state of art. These are only examples, the introduction requires some work.
  2. Figure 2a. I do not understand what is the point of these arrows with the designation “tool rotation speed”, “welding speed”. What they are pointing? Is this figure correct?
  3. Figure 3 presents the joint containing a lot of flaws and I would like to see other macrostructure images of the examined welds. If they also contain imperfections it is difficult to conclude about IMC influence… And the grain microstructure has not been revealed, what can we see are just precipitates in the alloy structure and due to its cast nature the vast majority of them are on the grain boundaries, but it is difficult to discuss grain size based on that, especially in the SZ where the precipitates undergo fragmentation.
  4. Lines 331-350 in major part contains no discussion, it is only a report of what we can see in Fig.4.
  5. I have some serious doubts about the methodology of IMC layer thickness measurements. How many samples have been tested and from what part of the joint they have been taken?
  6. The entire paper is very scruffy… as an example, we can see the graph in figure 7. The legend is messy, there are some brackets with no purpose, also I have checked a reference [16] and the authors should add the reference to figure 3 (and 1) for it has been also taken from the previous work. If this is how the joints look like and would not bother with IMC…
  7. No explanation of the shear test and sample geometry is given.
  8. The authors focused on IMC thickness completely neglecting: joints’ imperfection (Fig. 3., Figure 9 from [16]) and cracks in IMC layers (e.g. Fig. 5. c). For this reason, it is not a good research work and I can not recommend the publication.

Overall: reject.

Author Response

Dear reviewer

Thank you for your kind and keen comments.

We will expect your favorable evaluation.

Best regards,

Leejon Choy

Reviewer 2 Report

Article “Effect of Microstructure and Tensile Shear Load Characteristics Evaluated by Process Parameters in Friction Stir Lap Welding of Aluminum-Steel with Pipe Shapes” (revised)

As for the relevance of the article, it is beyond doubt. Obtaining heterogeneous welded joints of aluminum alloys to steels is a very urgent task for a number of elements in the automotive industry, mechanical engineering, and power engineering. But it is often difficult to ensure high quality of these welded joints - brittle intermetallic interlayers that form in welded joints obtained by fusion welding lead to a sharp drop in their operational properties. In this regard, most technologies for welding dissimilar elements from aluminum alloys to steels do not find real practical application. The use of friction stir welding, in which less favorable conditions for the interlayers formation are realized, is a more promising technology for solving such problems.

The introduction in the revised version of the article has been finalized. A detailed explanation of the subject under the study was added.

In the section on research methods (section 2), the quality of the graphic material was improved, and the terms were clarified.

However, the article has some weaknesses:

  1. In the introduction (lines 296…297) it is stated that the influence of TPD on IMC thickness and the quality of the welded joint is the main subject of the study. But in the article itself, the authors consider only three TPD values (0; 0.5 and 1 mm). This is not enough to give unambiguous and reliable conclusions. In general, as I wrote earlier, the matrix of the experiment is poor.
  2. Figure 3a contains a photograph of the welded joint structure, on which defects are visible. Of course, such defects are unacceptable in welded joints, and the joints must be rejected without mechanical testing. This joint may have good static strength, but its’ actual operational characteristics will be very poor, especially when used under dynamic or shock loading. Are there any photographs of better quality welds, which can explain the structure of the zones of the welded joint? Or is this joint the best among all joints obtained by the authors?
  3. Based on the data presented in Table 3, it is difficult to draw an clear conclusion about the existence of a relationship between TSL and IMC thickness. Most likely, this is due either to the presence of defects in the welded joints, or insufficient statistics. Or both reasons.
  4. To begin the article's conclusion by saying that the purpose of the article was not to prevent defects (line 1108) is, I think, incorrectly. The same applies to the introduction (lines 340…343).

Minor remarks:

WS (line 77) is not spelled out in the abstract.

“…the microstructure of the macrostructure…” (line 569). I don't think it's possible to write like that.

In general, of course, it is possible to publish an article, but the article should definitely be checked in detail by the authors. At the same time, the practical value of such an article is not obvious. I doubt that a ready-made technology has been obtained in the paper, which can be implemented in practice for the studied elements, which were mentioned in the introduction (near line 160).

Author Response

(The authors gave the same response as above.)

Round 2

Reviewer 1 Report

Thank you for your answers. I recommend the publication.

Author Response

Dear reviewer

We modified the misspelled expressin of our article.  

We deleted lines 155-159 and 602-604 regarding the text that defect elimination is not our purpose

We revised  duplicate sentences from the previous article of introduction

We expect a favorable reply ASAP.

Best regards,

Leejon Choy

This manuscript is a resubmission of an earlier submission. The following is a list of the peer review reports and author responses from that submission.

Round 1

Reviewer 1 Report

  1. The Introduction needs to be more detailedly provided, especially, the recent progress in this field should be logically presented.
  2. The mechanical properties of the base materials used in this study should be given in the paper.
  3. In Table 2, the welding speed (rpm) and tool speed (rpm) are difficult to understand, the names of process parameters should be well defined.
  4. According to the grouping of experiment 1-6 in Table 2, it can be found that there are two or more variables in the group A and B, thus it is difficult to investigate the effect of a specific welding parameter on the IMCs thickness and TSL.
  5. In Figure 3, the image scale should be given if necessary.
  6. Fig. 3 gives the cross section of the joint with void defect. The microstructures may be not representative for this study.
  7. The conclusions should be condensed to highlight the new findings of the paper. 
  8. The novelty of the paper was also not well provided in the abstract. The abstract should be rewritten to highlight in a compact way the background, method, new results and new findings (or new mechanisms), and the significance of the study.
  9. The paper mainly focused the relations between process parameters and IMC features, however, the authors claimed that “These results indicate that the IMC thickness is not the dominant factor that determines the magnitude of the TSL (lines 609-610)”. For such a case, I think the investigation on IMC is of little significance.
  10. What is the intrinsic reason for the influence of parameters on the IMC features? The current manuscript looks like an experiment report. The deep discussion of the experiment results is lack.
  11. Which are the most important factors that affect the mechanical properties of the joints? Why?
  12. Reference format needs to be modified according to the submission requirement.

Author Response

Dear Reviewer

Thanks for kind review.

I'd like to acept ASAP.

Best regards,

Leejon Choy

Reviewer 2 Report

The article “Effect of Microstructure and Tensile Shear Load Characteristics Evaluated by Process Parameters in Friction Stir Lap Welding of Aluminum-Steel with Pipe Shapes” is relevant.

The production of dissimilar welded joints of aluminum alloys to iron-based alloys is in demand in the industry, mainly for the purpose of reducing the structures weight. At the same time, it is difficult to obtain a high-quality welded joint of aluminum alloy to steel using traditional fusion welding technologies. It is enough to look at the “Fe-Al” phase diagram to see the amount of FexAly intermetallics that will inevitably be present in the fusion weld. Taking into account the fact that most of these intermetallics are brittle, their presence in the welded joint greatly reduces its structural strength. As a result, friction stir welding (FSW) is currently a worthy alternative to traditional welding technology and is a promising method for obtaining dissimilar welded joints of the required quality.

The article discusses one of the welded joints of cast aluminum alloy and HSS used in the automotive industry. The problem is private, but the results obtained could potentially be useful for other products made from these (or similar) alloys.

In general, the research is experimental, and the article itself contains valuable results. At the same time, the present study has a number of issues that require adjustments and appropriate explanations.

The main point is that the approach to research is not detailed enough:

- The article does not analyze the reasons for the influence of welding mode parameters on the IMC thickness. It is simply said that, for example, that "with an increase in this parameter, the thickness of the IMC decreases." No explanations or even assumptions are presented why this happens. 

- Attention in the paper is rightly paid to the IMC thickness, which greatly affects the properties of the welded joint. However, no data are provided on the chemical composition of this layer, the gradient of the chemical composition along the IMC thickness. Analysis of the IMC chemical composition and the dynamics of its change depending on the welding mode would make it possible to obtain a more accurate direction for improving the welding mode.

- The authors mention the poor mechanical properties of welded joints with large IMC. However, in the paper itself, no studies of the mechanical properties of welded joints and its zones were carried out. The microstructure of all welded joint zones has been investigated, but even the microhardness of the metal in these zones has not been estimated. Microhardness studies could indicate weak and dangerous zones of welded joints and show the welding modes at which the size and degree of danger of such zones would be minimal.

In my opinion, the authors have not considered the study of welded joints in sufficient detail (even from the point of view of the methods used). Because of this, and also because of the total small number of modes studied (six modes), the results obtained in this paper are not obvious. The graphs shown in Figures 7-10, and especially in Fig. 7 and Fig. 10, look unconvincing - there is no clear influence of the FSW mode parameters on TSL and IMC due to the small number of experiments.

It is obvious that the value of this article could lie not in new scientific knowledge, but in good practical recommendations, but looking at the results obtained, it is difficult to see ready-made technological results that can be applied in practice to obtain welded joints of the studied alloys.

In addition, there are a couple of private remarks:

  1. I suppose, it is necessary to give an explanation to Figure 1c. There is no reference to it in the text in the paragraph before the figure 1, and upon first reading it is not immediately clear what exactly is shown in the figure 1c and why the TPD was varied.
  2. Perhaps it is worth clarifying in more detail how the TSL was assessed? Methods, equipment, tooling. Why are there three curves in the Figure 2d are shown?

The authors are invited to supplement and modify the article. I believe that after sufficient revision, the article can be considered for publication in "Materials".

Author Response

(The authors gave the same response as above.)

Reviewer 3 Report

The paper presents a value for industrial applications. I highly recommend to improve a lot of issues before its publication.

  1. You have to check the language in the entire paper, in some places, it is very odd and hard to understand. Three examples only from one section: “grains are increased “ (Line 251), “magnitude of the welding parameters” (Line 257), “healthy weld” (Line 258).
  2. The images “c” and “d” in Fig. 2 are of poor quality. The scale in 2c is impossible to read.
  3. Poor choice of representative weld for Fig. 3. It has 1 mm void in the central part of the weld and I have some serious doubts about the coherency between steel and aluminum in this joint for its major part is delaminated. The formation of the void is not discussed. The scales should be improved and I have no idea what do you consider as the “plunge depth” in this case. This parameter does not correspond to the marked line (whatever it aims to represent). Advancing and retreating sides should be pointed in the macroscopic image. Also, the entire section “discussing” Fig. 3 consists of well-known features of FSW joints microstructure and does not refer to the actual results. Fig. 3b does not show any grains but fragmented precipitates due to bad etching. And for sure there are some IMC at the interface marked in Fig. 3c but this is impossible to conclude from the presented image.
  4. Lines 282-297 should belong to the introduction part, although it is relatively poor-written and contains well-known things (e.g. “High heat affects the atomic diffusion rate and microstructure” – line 287) and impossible to understand sentences (e.g. “The IMC thickness affects the strength of the machine (…) – line 296).
  5. How is it possible to construct the diagram (Fig. 6) based on single measurements of IMC thickness? It is unacceptable for a research paper. Where are error bars?
  6. Figure 7 – the same remarks as in the previous point, just change “IMC thickness” with “tensile shear force”. Additionally, where is the geometry of the sample for this test? How this test has been performed? How many samples have been tested for each state? Also, I recommend to present stress instead of force.
  7. I think that the visible imperfections, mostly cracks next to the IMC layer (e.g. Fig. 5c) are the crucial factor for the load-carry capacity of the joints. You do not pay any attention to the imperfections, only focusing on IMC what, in my opinion, is a clear mistake in the analysis.
  8. IMC layer should be subjected to the analysis of chemical composition to show the distribution of the elements and clearly present that this layer is of IMC origin.
  9. Vast majority of this paper consists of well-known facts and simple reading of the diagrams. The discussion is weak and it should be improved.

Overall: major!

Author Response

(The authors gave the same response as above.)

Round 2

Reviewer 1 Report

The modifications have been well prepared and the paper can be accepted for publication.

Author Response

Dear reviewer

Thank you for your kind comment again.

I'd like to receive your favorable reply ASAP.

Best regards,

Leejon Choy

Reviewer 2 Report

“Effect of Microstructure and Tensile Shear Load Characteristics Evaluated by Process Parameters in Friction Stir Lap Welding of Aluminum-Steel with Pipe Shapes”.

Round 2 review.

The article was corrected by the authors, and most of the recommendations were partially or fully taken into account.

+ Brief explanations are given about the influence of welding mode parameters on the IMC thickness.

+ The cover letter gives brief information about the IMC chemical composition (EDS analysis results are performed). Some information explaining the mechanical tests carried out is shown.

+ Explanations regarding the scientific novelty and relevance of the paper are provided.

+ Particular remarks on Figure 1 and the TSL test procedure were corrected.

+ The introduction and conclusions have also been corrected in the positive way.

- Still I think that the representation of the influence of welding mode parameters on TSL results and IMC is not good enough. I propose to plan more carefully the matrix of experiments in subsequent studies. The experiment matrix should be more detailed when investigating the effect of several welding mode parameters.

In general, the article looks better now. I consider that publication in the "Materials" is possible.

Author Response

(The authors gave the same response as above.)

Reviewer 3 Report

The paper still requires a lot of work. If Authors would like to get “acept ASAP.” (original spelling from author’s response – unbelievable…) I suggest to put more effort into this manuscript.

  1. Language issue. The authors need to fix the language in the entire paper not only in the three examples given by me. This time only one example: “more intense texture” (line 445)…?
  2. Some parts of the paper are still unacceptable for research work. E.g. lines 412-422 (in the new version of the manuscript) are simple text-filler that provides no additional information to the “discussed” figure.
  3. I do not accept the author’s answer for comment “3-1”. The authors aim to investigate (among others) the relation between IMC thickness and shear strength and in my opinion, this is pointless when the test is performed on joints with such serious defects. Please, discuss it. And what does it mean “It is not intended for optimization by increasing weldability”?
  4. I also do not accept the author’s answer for comment “3-2”. The authors wrote, “A flash occurred during the experiment, resulting in large voids due to insufficient material to fill the RS area behind the tool”. First, the flash does not explain such a large void, I can show pictures from my research with far larger flash with defect-free joint and also void-having joint with no flash. So I do not accept this explanation. Secondly, in the response sent by authors, I have an opportunity to see different welds performed by them and in the major part there is no flash (or it is relatively small) and the voids occurred anyway.
  5. Authors still do not know what is “plunge depth”. It is measured from the top of the pin and not from the shoulder. When a pin is placed directly on the surface of the workpiece (without penetration) the plunge depth is “0”. So if the workpiece is 3 mm thick and there is a complete penetration, the plunge depth has to be greater than 3 mm. It is an important parameter, crucial for the repeatability of the research so it has to be given correctly.
  6. “Our response (5) This data guarantees the reliability of the data as a result of three or more experiments for the same level through the design of experiment method.” And the result is the same? For each of these measurements? No error bars? I can see the “IMC thickness range” in table 3, is it impossible to calculate standard deviation?
  7. “Our response (6) This data guarantees the reliability of the data as a result of three or more experiments for the same level through the design of experiment method. The geometry of the sample for this test” – so why the authors did not put this scheme in the paper? I can see that no explanation about this test has been added to the experimental part. Unbelievable. And no error bars… what for? “This data guarantees the reliability of the data”.
  8. “Our response (7) Thank you for opinion, figure 5c is not a welding defect, but a micro-deformation caused by an impact during hot mounting for the creation of a microstructure specimen. There were no cracks in the specimen before impact” – How do authors know? What observations have been performed before the metallographic preparation? What was the temperature of mounting? I do not accept this answer at all.

I am deeply unsatisfied with the given answers and with the fact that the authors did not fix the manuscript according to suggestions.

I can not recommend the publication.

Author Response

(The authors gave the same response as above.)
